# New Pyrimidine-5-Carbonitriles as COX-2 Inhibitors: Design, Synthesis, Anticancer Screening, Molecular Docking, and In Silico ADME Profile Studies

**DOI:** 10.3390/molecules27217485

**Published:** 2022-11-02

**Authors:** Hanan A. AL-Ghulikah, Samiha A. El-Sebaey, Amr K. A. Bass, Mona S. El-Zoghbi

**Affiliations:** 1Department of Chemistry, College of Science, Princess Nourah Bint Abdulrahman University, P.O. Box 84428, Riyadh 11671, Saudi Arabia; 2Department of Pharmaceutical Organic Chemistry, Faculty of Pharmacy (Girls), Al-Azhar University, Cairo 11754, Egypt; 3Department of Pharmaceutical Chemistry, Faculty of Pharmacy, Menoufia University, Shibin-Elkom 32632, Egypt

**Keywords:** COX-2 inhibitiors, cyanopyrimidine, anticancer, benzoazoles, benzenesulphonamides

## Abstract

Two series of cyanopyrimidine hybrids were synthesized bearing either benzo[*d*]imidazole, benzo[*d*]oxazole, benzo[*d*]thiazole, and benzo[*b*]thiophene derivatives via methylene amino linker **3a**–**3d** (Formula A) or various sulphonamide phenyl moieties **5a**–**5d** (Formula B) at the C-2 position. All compounds’ cyclooxygenase COX-2 inhibitory activities were evaluated, and all synthesized compounds demonstrated potent activity at minimal concentrations, with IC_50_ values in the submicromolar range. Compounds **3b**, **5b**, and **5d** were discovered to be the most active pyrimidine derivatives, with the highest COX-2 percent inhibition and IC_50_ values being nearly equal to Celecoxib and approximately 4.7-, 9.3-, and 10.5-fold higher than Nimesulide. Furthermore, the pyrimidine derivatives **3b**, **5b**, and **5d** demonstrated anticancer activity comparable to or better than doxorubicin against four cell lines, i.e., MCF-7, A549, A498, and HepG2, with IC_50_ values in nanomolar in addition to low cytotoxicity on the normal W38-I cell line. The effect of compound **5d** on cell cycle progression and apoptosis induction was investigated, and it was found that compound **5d** could seize cell growth at the sub-G1 and G2/M phases, as well as increase the proportion of early and late apoptotic rates in MCF-7 cells by nearly 13- and 60-fold, respectively. Moreover, in silico studies for compounds **3b**, **5b**, and **5d** revealed promising findings, such as strong GIT absorption, absence of BBB permeability, nil-to-low drug–drug interactions, good oral bioavailability, and optimal physicochemical properties, indicating their potential as promising therapeutic candidates.

## 1. Introduction

Cyclooxygenases (COXs), generally recognized as prostaglandin-endoperoxide synthase (PTGS), are members of the isozymes family, which is responsible for catalyzing arachidonic and polyunsaturated fatty acids to prostaglandin PGE2, PGD2, PGI2, and PGF2α, as well as thromboxane A2 (TXA2), by following the arachidonic acid pathway [1,2]. At present, three isoforms of the COX enzyme, i.e., 1, 2, and 3, have been documented [3,4]. Among these isoforms, cyclooxygenase-2 (COX-2) is a pro-inflammatory enzyme driven by cytokines or lipopolysaccharide (LPS) and is interrelated with angiogenesis, inflammation, and cancer [5,6]. Mostly, the COX-2 enzyme is the source of carcinogenesis in several types of cancers and plays a role in cancer cell resistance in chemo- and radiotherapies. COX-2 may also cause the activation of matrix metalloproteinase, which follows cell membrane degradation, metastasis, and tumor invasiveness [7,8]. It was reported that COX-2 displays many cancer-causing activities, including apoptosis inhibition, EGF receptor stimulation, and tyrosine [9,10]. Furthermore, COX-2 was found to be responsible for the beginning of P-glycoprotein manufacturing, which is over-expressed in drug-resistant tumors [11]. However, under normal circumstances, COX-2 plays a critical role in renal function regulation in both normal and renal insufficiency, liver cirrhosis, and congestive heart failure illnesses [12]. Conversely, COX-2 levels rise in the condition of inflammation and/or cancer, which leads to the discharge of huge amounts of inflammatory prostaglandins [13]. Therefore, COX-2 inhibition has somehow become important in reducing COX-2-linked activities that lead to a decline in the appearance and development of tumors, apoptosis augmentation, and metastasis of cancer [14].

The pyrimidine nucleus is among the major structural cores in medicinal chemistry, because it produces a variety of therapeutically useful medicines. It is indicated from the literature that pyrimidine and its derivative compounds hold kinase- and angiogenesis-inhibition potential [15]. Most pyrimidine-fused enzyme inhibitors combined with DNA or RNA cause inhibition and misinterpretation of DNA polymerase [16,17]. Further, pyrimidines possess the capability of acting as anticancer, antioxidant, and COX-2-inhibiting agents [18,19]. Hence, designing multi-target compounds mainly composed of pyrimidine may be beneficial in treating many cancer-related diseases instead of conventional drugs. In addition, pyrimidine inhibitors of COX-2 have significant strength in managing disorders by suppressing angiogenesis and inflammation [20].

### The Rationale of Molecular Design

The pyrimidine core is the primary structural skeleton of many chemotherapeutics used for cancer treatment, including fluorouracil, ribociclib, brigatinib, roscovitine, etravirine, risperidone, iclaprim, avanafil, and rosuvastatin [21]. On the other hand, cyano-bearing drugs such as anastrozole, bicalutamide, and apalutamide are effective anti-androgens used in the treatment of breast and prostate cancer [22]. Further, some researchers have designed anticancer compounds containing a pyrimidine nucleus with a cyano moiety at the C-2 and C-5 positions; see Figure 1. For instance, Shao et al. synthesized 2,5,6-trisubstituted cyanopyrimidine analogs with secondary amine derivatives at the C2 position, compound **I** of which provided strong anti-proliferative efficacy against MCF-7 and HCT-116 (GI_50_ = 0.64 µM, 0.79 µM) [23]. Based on a molecular docking simulation investigation, Fathalla et al. synthesized a sequence of pyrimidine-benzene sulfonamide derivatives with a secondary amine moiety at C-2. Of all of the derivatives, compound **II** showed good anticancer activity in vitro toward HeLa cell lines, with an IC_50_ = 0.039 µM [24]. Similarly, compounds **III** and **IV** with a cyanopyrimidine ring, in which the -NH group was substituted with isosteric sulphur and methylene at C-2, were discovered to be strong anticancer agents against the majority of the established subpanel tumor cell lines, particularly gastric MGC-803 cancer cells, with IC_50s_ = 4.01 and 5.85 µM, respectively [25,26]. On the other hand, 6-(4-methoxyphenyl)-dihydropyrimidine-5-carbonitrile derivative **V** has demonstrated significant COX-2 selectivity and also has a substantial effect on MDA-MB-231 (IC_50_ = 3.43 µM) and MCF-7 (IC_50_ = 2.56 µM) breast cancer cell lines [27]. Compound **VI**, which has a pyrimidine ring with cyano and *p*-methoxyphenyl moieties at the C-5 and C-6 positions, showed outstanding anticancer activity against most of the NCI60 cell lines, in addition to showing promising COX-2 inhibitory activities [28].

The benzenesulfonamide moiety constitutes an essential feature for selective COX-2 inhibitors [29] as well as the commercial COX-2 inhibitors drugs, such as celecoxib, apricoxib, and parecoxib. Additionally, compound **VII** (Figure 2), which contains a pyrimidine-based sulfonamide phenyl pharmacophore, outperformed rofecoxib (IC_50_ = 292 nM) by 780-fold (IC_50_ = 12.7 nM) in COX-2 inhibitor potency [30]. Furthermore, a N-(2,6-dimethoxypyrimidin-4-yl) benzenesulfonamide derivative **VIII** (Figure 2) inhibited COX-2, with IC_50_ values in the range of 0.22–0.67 µM, which was higher than Celecoxib (IC_50_ = 7.70 µM). Such compounds have demonstrated promising anticancer activity against a variety of cell lines, indicating a connection between selective anti-COX-2 effects and their anti-cancer activity [31]. In addition, benzoazoles (purine analogs in general) constitute a significant pharmacophore in both anticancer agents and COX-2 inhibitors; for example, rutecarpine, which has an indole moiety, is a potent COX-2 inhibitor with no adverse effects on the cardiovascular system. Further, benzoxazole and/or benzimidazole hybrid structures with a pyrimidine scaffold, such as compound **IX** (Figure 2), were found to significantly inhibit COX-2 enzymes. These synthesized derivatives showed activity against breast carcinoma (MCF-7) (IC_50_ = 5.4 to 7.2 µM) and non-small-cell lung cancer (A549) (IC_50_ = 8.4 to 9.2 µM) [32]. Likewise, compound **X** (Figure 2), consisting of a pyrimidine scaffold and benzo[*c*] [1,2,5]oxadiazole and/or benzo[*c*] [1,2]oxazole moieties, inhibited COX-2 (IC_50_ = 4.6 µM), with moderate anticancer activity against human colon cancer cells. Furthermore, the presence of benzo[*d*]thiazole displayed a high COX-2 inhibitory effect (IC_50_ = 5.0 μM) [33]. Enlightened by the above-mentioned fundamentals, we developed a ‘hybrid model’ scaffold consisting of two series of cyanopyrimidines with *p*-methoxyphenyl moiety as pharmacophores that hybridize with either benzo[*d*]imidazole, benzo[*d*]oxazole, benzo[*d*]thiazole, and benzo[*b*]thiophene derivatives at the C-2 position **3a**–**3d** (Formula A) via a methylene amino linker or various sulphonamide phenyl moieties **5a**–**5d** (Formula B). The target compounds were subjected to anti-COX-2 activity screening, and the most active compounds were further evaluated for their cytotoxicity in order to develop potent COX-2 inhibitors with good anticancer activity that might be used in cancer treatment. The structure of cyanopyrimidine-based COX-2 inhibitors (Figure 1) and the rationale for designing the target compounds to potentiate the anticancer and COX-2 inhibitory activities in this study are given in Figure 2.

## 2. Results and Discussions

### 2.1. Structural Elucidation

A solvent-free protocol was designed to synthesize the compounds **3a**–**3d** and **5a**–**5d** by utilizing the starting substrate 2-amino-4-methoxy-6-(4-methoxyphenyl)- pyrimidine-5-carbonitrile **(1)**, which was synthesized from 2-(methoxy(4-methoxyphenyl) methylene)malononitrile and cyanamide according to the reported procedure [34]. Briefly, reagents with the substrate compound were thoroughly ground and/or mixed at room temperature and subjected to fusion at a suitable temperature. After that, mixtures were triturated with ethanol, acidified with dilute HCl, poured onto crushed ice, and crystallized with a suitable solvent. When compared to previous methods, the current procedure is much greener for the synthesis of pyrimidine derivatives due to the solvent-free conditions, easily obtainable resources, and high yields with facile product isolation. The preparation of the target compounds **3a**–**3d** (general formula A) and **5a**–**5d** (general formula B) is demonstrated in Figure 1 and Figure 2.

The importance of benzimidazole, benzoxazole, and/or benzothiazoles (purine analogs in general) inclues their renown for pharmacophoric scaffolds in cancer treatment and a selective inhibition of the COX-2 enzyme. Therefore, we have attempted to enhance the cytotoxicity and anti-COX-2 activity by the synthesis of pyrimidines bearing these motifs at the C-2 position through a methylamino linker **3a**–**3d** (general formula A) using a new synthetic methodology, Figure 1. Compounds **3a**–**3d** were synthesized through the condensation of compound **1** under a solvent-free technique with 2-(chloromethyl)-1*H*-benzo[*d*]imidazole, 2-(chloromethyl)benzo[*d*]oxazole, 2-(chloromethyl)benzo[*d*]thiazole, and (benzo[*b*]thiophen-2-yl)methanol, respectively, producing the target compounds in good yield (69–83%) via the elimination of HCl and/or water molecules. The structures of these compounds were confirmed by the existence of ^1^H NMR singlet signals at δ 4.02–4.73 ppm attributed to two protons of the methylene group of methylamino (-NHCH_2_-) fragments and deuterium oxide-exchangeable signals at δ 12.36–12.97 ppm, verifying the formation of an NH proton. Concurrently, the ^13^C NMR spectrum revealed signals at δ 60.24–61.56 ppm correlated to the methylamino carbon atom. The remaining expected proton and carbon signals were detected in the ^1^H NMR and ^13^C NMR spectra and are attributed in the experimental section.

As illustrated in Figure 2, a new series of pyrimidine derivatives bearing different sulphonamide phenyl moieties at the C-2 position **5a**–**5d** (general formula B) has been synthesized using the starting compound **1** as a building block via versatile and straightforward synthetic routes. The fusion of compound **1** at an equal molar concentration ratio with different sulfonyl chlorides, namely, benzenesulfonyl chloride, 4-methylbenzene-1-sulfonyl chloride, 4-*tert*-butylbenzene-1-sulfonyl chloride, and 4-(trifluoromethyl)benzene-1-sulfonyl chloride at 200 °C, resulted in the synthesis of the respective analogs **5a**–**5d** in 72–81% overall yields.

The chemical structures of compounds **5a**–**5d** were established with the help of spectral and elemental examinations. The structure of compound **5b** was confirmed by the appearance of a singlet at 2.29 ppm attributed to methyl protons in the ^1^H NMR spectrum and a signal due to methyl carbon at 23.53 ppm in the ^13^C NMR spectrum. As for compound **5c**, the ^1^H NMR and ^13^C NMR spectra exhibited the presence of a singlet signal corresponding to the *tert*-butyl moiety at 1.28 ppm and 31.92 ppm, respectively. The formation of compound **5d** was verified by the appearance of two doublet signals at 6.54 and 6.93 ppm in the ^1^H NMR spectrum, which is characteristic of 4-(trifluoromethyl)benzene sulphonamide protons. In addition, the ^13^C NMR spectrum revealed a signal assignable to CF_3_ carbon at 120.98 ppm. The obtained results of all the synthesized compounds are in good agreement with the estimated values for the proposed structure, and the MPs are sharp, suggesting the synthesized compound’s purity.

### 2.2. Biological Evaluation

#### 2.2.1. In-Vitro COX-2 Inhibition

The results of COX-2 inhibition by synthesized compounds are given in Table 1. The pyrimidine derivatives, along with reference drugs Celecoxib and Nimesulide, were prepared in the range from 10^−5^ M to 10^−9^ M concentrations. Therefore, the IC_50_ values of the pyrimidine derivatives and reference drugs’ findings are comparable for validation. The tested compounds displayed more than 50% inhibitory activity of COX-2 at 10^−8^ M and 10^−9^ M concentrations. All compounds displayed a remarkable inhibition activity at lower concentrations (10^−8^ M to 10^−9^ M). However, the pyrimidine substituted with various sulphonamide phenyl derivatives **5a**–**5d** (general formula B) showed higher inhibition of COX-2 enzyme activity than the pyrimidines bearing benzimidazole, benzoxazole, benzothiazole, and benzothiophene derivatives **3a**–**3d** (general formula A). Notably, all pyrimidine derivatives exhibit significant inhibitory activity against COX-2 at a lower dosage, and the electron-withdrawing groups seem to be more active; however, the COX-2 enzyme appears to take in a wide range of electron-donating (CH_3_, OCH_3_) and electron-withdrawing (F) groups. Figure 3 depicts the SAR of the target compounds. On the other hand, **3a** and **3c** demonstrated a similar pattern of COX-2 inhibition as reference drugs. Among the selected pyrimidine derivatives, compounds **3b, 5b,** and **5d** at 10^−8^ M concentration were found to be the most active agents with the highest COX-2 % inhibition (77.01 ± 0.03, 75.25 ± 1.1, and 76.14 ± 1.05) and IC_50_ value (0.20 ± 0.01, 0.18 ± 0.01, and 0.16 ± 0.01 µM). Compounds **3b** and **5b** exhibited inhibitory action that was nearly comparable to Celecoxib, but they were about 4.7-and 9.3-fold more potent than Nimesulide, respectively. Further, it was observed that compound **5d** displayed a more potent inhibition profile of at least more than 10-fold in comparison to Nimesulide (IC_50_ 1.68 ± 0.22 µM) and a similar inhibition profile as Celecoxib (IC_50_ 0.17 ± 0.01 µM). Similar findings were reported when an ^18^F-labeled pyrimidine scaffold showed selective COX-2 inhibition [35]. Therefore, we could conclude that compounds **3b**, **5b**, and **5d** are the most active pyrimidine derivatives against the COX-2 enzyme.

#### 2.2.2. Anticancer Screening (MTT Assay)

In vitro anticancer screening of the most active compounds, **3b**, **5b**, and **5d**, against non-small-cell lung cancer (A549), breast carcinoma (MCF-7), renal carcinoma (A498), and liver carcinoma (HepG2) cell lines were assessed by the MTT assay (Table 2). All three compounds, along with the reference drug, doxorubicin, were active in this cell line. From the obtained data, the three compounds (**3b**, **5b**, and **5d**) demonstrated potent cytotoxic activity against all cell lines in nanomolar that were 2- to 13-fold higher than doxorubicin, except for compound **3b**, which demonstrated anticancer activity against A549 (IC_50_ = 19 ± 0.52 nM) and HepG2 (IC_50_ = 22 ± 0.62 nM) comparable to doxorubicin (IC_50_ = 13 ± 0.42 and 25 ± 0.55 nM, respectively). It is worth noting that compound **3b** displayed three- and two-fold stronger anticancer activity than doxorubicin against the MCF-7 and A498 cell lines, respectively (see Table 2). Furthermore, compound **5b** was demonstrated to be 4.5-, 4.3-, 7-, and 2.1-fold more cytotoxic than the reference drug against the three MCF7, A549, A498, and HepG2 cell lines, respectively. Moreover, compound **5d** was the most potent agent, since it showed a higher potency than doxorubicin against the MCF7, A549, A498, and HepG2 cell lines by about 9-, 13-, 3.5, and 2.8-fold, respectively.

To assess the safety of the synthesized compounds, **3b**, **5b**, and **5d** were investigated against the non-tumorigenic WI-38 cell line (normal cell derived from lung tissue). The results indicated that the tested compounds had low cellular cytotoxicity, with high IC_50_ values ranging from 89.62 ± 2.75 to 94.71 ± 2.5 µM in contrast to doxorubicin, which showed high cellular cytotoxicity with a low IC_50_ value of 8.83 ± 0.09 µM, as shown in Table 2.

#### 2.2.3. Cell Cycle Analysis

The most active compound, **5d**, was selected for further investigation regarding its effect on cell cycle progression and induction of apoptosis in the MCF-7 cell line. When MCF-7 cells were treated with compound **5d** at its IC_50_ concentration, the cell population accumulated significantly in both the sub-G1 and G2/M phases (by 16 and 5 times, respectively) when compared to the control. In addition, S-phase cells showed a significantly lower DNA content percentage (16.23%) than control cells (29.02%) (Table 3 and Figure 4). These findings imply that compound **5d’s** cytotoxicity was proven by the arrest of cell growth at the G2/M phase.

#### 2.2.4. Apoptosis Determination by Annexin-V Assay

A biparametric cytofluorimetric analysis was performed using the Annexin-V assay to verify the ability of compound **5d** at its IC_50_ concentration to trigger apoptosis. When compared to the control, compound **5d** increased the proportion of early and late apoptotic cells in MCF-7 cells by nearly 13- and 60-fold, respectively, when compared to control cells, which had 0.53% and 0.25% early and late apoptotic cells, respectively. Furthermore, compound **5d** had a three-fold higher proportion of necrosis than the control, as demonstrated in Table 4 and visually in Figure 5.

### 2.3. In Silico Studies

#### 2.3.1. Docking Studies of Compounds **3b**, **5b** and **5d**

A molecular modeling study on the most active compounds (**3b**, **5b**, and **5d**) was performed on the cyclooxygenase-2 enzyme to examine the binding modes of the target compounds inside the pocket of the cyclooxygenase-2 enzyme. This study was conducted using MOE 2014 software on the COX-2 enzyme and compared the docking results of these compounds with the binding mode of Celecoxib as the reference compound in an attempt to rationalize their mechanism as anti-inflammatory agents. For this study, the 3D crystal structure of the COX-2 (PDB ID: 1CX2) complex with the reference Celecoxib was chosen, which was retrieved from the PDB bank (http://www.rscb.org/pdb)(accessed on 15 March 2022). To validate the docking steps, the Celecoxib co-crystallized conformer was re-docked into the COX-2 binding site, and the docking pose was compared with the initial pose utilizing root mean square deviation (RMSD = 0.699 Å). Celecoxib binds to the COX-2 isoform, forming three hydrogen-bonding interactions with the amino acid residues Leu352, Gln192, and Arg513 and one arene-H interaction with the amino acid residue Ser353, with a binding affinity of −6.13 kcal/mol. Docking of compound **3b** within the COX-2 enzyme showed a hydrogen-bonding interaction between methylene hydrogen and the same amino acid residue, Leu352 (3.05 Å), as the reference Celecoxib, in addition to an arene–cation interaction between the *p*-methoxy phenyl moiety and the amino acid residue Arg120 (4.29 Å) with an energy score = −5.64 kcal/mol. Moreover, docking of compound **5b** revealed two hydrogen-bonding interactions with the same amino acid residues, Leu352 (2.94 Å) and Arg513 (3.21 Å), as the reference drug, with an energy score = −4.93 kcal/mol. Compound **5d** fit well with the binding site, as the ligand formed three hydrogen-bonding interactions with the amino acid residues Leu352 (3.10 Å), His90 (3.24 Å), and Arg513 (2.68 Å), in addition to an arene–cation interaction between the *p*-methoxy phenyl moiety and the amino acid residue Arg120 (4.19 Å), with an energy score = −3.76 kcal/mol, as illustrated in Figure 6andFigure 7. The docking data of the synthesized compounds **3b**, **5b**, and **5d**, as well as Celecoxib in COX-2 active sites, were depicted in Table 5.

#### 2.3.2. In Silico Physicochemical Features, Pharmacokinetics Profiles, and Drug-Likeness Data of **3b**, **5b**, and **5d** Compared to Celecoxib

To be a potential drug candidate, molecules must possess certain features, i.e., pharmacokinetics or pharmacodynamics, as well as physicochemical features and drug-likeness. Therefore, to study the silico ADME profile of the compounds **3b**, **5b**, and **5d** in comparison with Celecoxib, Swiss ADME online software (www.SwissADME.ch) was utilized. The Boiled-Egg chart [36] revealed that compounds **3b** and **5d** are likely to be absorbed more in the gastrointestinal tract (GIT), in a manner similar to Celecoxib, attributable to the latter’s position in the human intestinal absorption (HIA) region. However, compound **5b** was beyond the HIA zone, which indicates that its low ability to become absorbed in the GIT could delay oral bioavailability.

Additionally, the target compounds **3b**, **5b**, and **5d** were characterized by BBB permeability absence, like Celecoxib, suggesting that they might not enter the central nervous system (CNS) (Figure 8). Further, each compound (**3b**, **5b**, and **5d**) is anticipated to inhibit approximately three cytochrome P−450 (CYP) isoforms in the liver. Hence, to evade any possible drug–drug interactions, these studied compounds should be administered at time intervals when any other medication is also prescribed. Celecoxib was found to inhibit two CYP isoforms, CYP2C9 and CYP1A2 (Table 6).

Moreover, the oral bioavailability of the compounds **3b**, **5b**, and **5d**, as well as Celecoxib was also evaluated and is represented in Figure 9. The radar chart showed the vital oral bioavailability parameters, which include POLAR (polarity), SIZE (size), INSATU (saturation), FLEX (flexibility), INSOLU (solubility), and LIPO (lipophilicity). The pink area in the radar chart indicates the ideal range for parameters, and the red lines represent the computed physicochemical properties of the material. The measured physicochemical values of the three compounds were found to reside in the pink region, with the exception of the INSATU parameter, which showed a violation identical to Celecoxib.

Table 7 lists the physicochemical parameters of compounds **3b**, **5b**, and **5d**. The target compounds have a molecular weight < 500 Da, which complies with Lipinski’s rule, suggesting that they will easily diffuse and absorb into the cell membrane. They are also expected to have high membrane permeability, since they meet the ideal log P-value. Furthermore, compounds **3b**, **5b**, and **5d** were shown to be ideal H-bond acceptors (7-10) and H-bond donors (1), allowing the molecule to pass through the aqueous pores of biological membranes by passive diffusion. Furthermore, each of these compounds contains six or seven rotatable bonds, each of which is connected to a heavy atom, implying good molecular flexibility. The polar atoms of the molecule, alternatively, produced moderate TPSA. According to NRB and TPSA, the target compounds exhibit an interesting oral bioavailability. In general, compounds **3b**, **5b**, and **5d** had physicochemical features that were nearly identical to Celecoxib (Table 7).

The SwissADME Web tool demonstrated that compounds **3b**, **5b**, and **5d** compiled most of the drug-likeness rules proposed by Ghose’s (Amgen) [37], Lipinski’s (Pfizer) [38], Veber’s (GSK) [39], Muegge’s (Bayer) [40], and Egan’s (Pharmacia) [41] filters for pharmaceutical companies. Lipinski and Veber’s rules are two of the most prominent methods for identifying drug-like compounds and their oral bioavailability. Lipinski’s guidelines focus on finding compounds with permeability and absorption problems, whereas Veber’s rules focus on molecular flexibility and topological polar surface area. Both Lipinski and Veber’s guidelines were entirely followed by the under-investigated compounds. The examined compounds do not have PAINS (Pan-Assay Interference Structures) or Breaks (Structural) alerts [42]. These results indicate that the **3b**, **5b**, and **5d** compounds lack interference in any protein test, which implies that the findings of the in vitro bioassays are reliable (Table 8).

## 3. Experimental

### 3.1. Materials

The starting compound (**1**), 2-amino-4-methoxy-6-(4-methoxyphenyl)pyrimidine-5-carbonitrile, was prepared in accordance with the reported method [34]. Other chemicals used in this study, i.e., 2-(chloromethyl)-1*H*-benzo[*d*]imidazole, 2-(chloromethyl) benzo[*d*]oxazole, 2-(chloromethyl) benzo[*d*]thiazole, (benzo[*b*]thiophen-2-yl)methanol, benzenesulfonyl chloride, 4-methylbenzene-1-sulfonyl chloride, 4-*tert*-butylbenzene-1-sulfonyl chloride, 4-(trifluoromethyl)benzene-1-sulfonyl chloride, ethanol, dimethylformamide, and hydrochloric acid (HCl) were purchased from the Merck, Sigma Aldrich, Invitrogen, and B.D.H. chemical companies. The organic solvent was used directly without any further purification.

### 3.2. Instrumentation

Synthesized compounds’ melting points (M.P) were assessed on the Electrothermal LA 9000 SERIES, digital MP apparatus. IR spectra were determined on a Shimadzu IR 435 spectrophotometer (KBr, cm^−1^). Elemental analyses of synthesized compounds were performed on the Elementar Vario EL III CHN analyzer. For structural elucidation of the synthesized compounds, ^1^H NMR (400 MHz) and ^13^C NMR (100 MHz) spectra were recorded on the Varian Gemini 400 MHz spectrometer. DMSO-d*_6_* was used as a solvent and TMS as an internal reference. Mass spectra were recorded on a Shimadzu Qp-2010 plus spectrometer. The purity of each compound was visualized under an ultraviolet irradiation lamp at λ 254 nm, which was established through TLC silica plates; TLC silica gel 60 GF254, Merck.

### 3.3. Procedure for Synthesis of Substrate Compound ***1*** [34]

Equimolar amounts (10 mmol) of 2-(methoxy(4-methoxyphenyl)methylene) malononitrile and cyanamide were heated under reflux in methanol with sodium methoxide (20 mmol) for 4h. The reaction mixture was allowed to cool before being poured into ice-cold water. The resulting solid was filtered, washed several times with water, left to dry, and crystallized from ethanol.

Compound **1** appeared to be yellow powder with 49% yield, having M.P. 195–197 °C. **IR** (**KBr**, **cm^−1^**): 3347, 3290 (NH), 3014 (C-H aromatic), 2980 (C-H aliphatic), 2210 (CN), 1635 (C=N), 1565 (C=C). **^1^H NMR (DMSO-*d_6_*, 400 MHz) δ:** 3.74 (s, 3H, OCH_3_); 3.82 (s, 3H, C_6_H_4_-OCH_3_); 6.88 (d, 2H, *J* = 7.55 Hz, C_6_H_4_-C_3,5_-H); 7.22 (d, 2H, *J* = 7.53 Hz, C_6_H_4_-C_2,6_-H); 8.21 (s, 2H, NH_2_, D_2_O exchangeable).

### 3.4. General Procedure for Synthesis of Compounds ***3a***–***3d***

An equimolar mixture of compound (**1**) and different reagents (3 mmol), namely, 2-(chloromethyl)-1*H*-benzo[*d*]imidazole, 2-(chloromethyl)benzo[*d*]oxazole, 2-(chloromethyl)benzo[*d*]thiazole, and (benzo[*b*]thiophen-2-yl)methanol were grinded and/or mixed well to become fused at ~200 °C for 3 h. Later, the mixture was cooled to become triturated with ethanol and acidified with dilute HCl. The formed components were left to dry after being filtrated and washed with ethanol. The resultant crude products were crystallized in a mixture of ethanol/dimethylformamide (3:1 *v*/*v*).

#### 3.4.1. 2-((1*H*-Benzo[d]imidazol-2-yl)methylamino)-4-methoxy-6-(4-methoxyphenyl)pyrimidine-5-carbonitrile (**3a**)

Compound **3a** appeared to be a light-brown powder with 78% yield, having M.P. > 300 °C. **IR** (**KBr**, **cm^−1^**): 3220(NH), 3062(C-H aromatic), 2966(C-H aliphatic), 2230 (CN), 1615(C=N), 1585 (C=C). **^1^H NMR (DMSO-*d_6_*, 400 MHz) δ:** 3.95 (s, 3H, C_6_H_4_-OCH_3_); 4.25 (s, 3H, pyrimidine-C_6_-OCH_3_); 4.73 (s, 2H, CH_2_); 7.38 (d, 2H, *J* = 6.84 Hz, C_6_H_4_-C_3,5_-H); 7.58–7.64 (m, 4H, C_6_H_4_-C_2,6_-H & benzo[*d*]imidazole-C_5,6_-H); 7.90 (d, 2H, *J* = 6.72 Hz, benzo[*d*]imidazole-C_4,7_-H); 12.81 (s, 1H, NH-CH_2_, D_2_O exchangeable); 12.93 (s, 1H, imidazole-NH, D_2_O exchangeable),(See Appendix A). **^13^C NMR (DMSO-*d_6_*, 100 MHz) δ:** 43.47 (CH_2_); 56.72, 57.13 (2OCH_3_); 80.05 (pyrimidine-C_5_); 114.96 (C_6_H_4_-C_3,5_); 118.42 (CN); 121.01, 124.89, 125.45 (benzo[*d*]imidazole-C_4,5,6,7_); 129.28, 129.34 (C_6_H_4_-C_2,6_); 138.41, 138.59 (benzo[*d*]imidazole-C_3a,7a_); 148.28 (benzo[*d*]imidazole-C_2_); 159.06 (C_6_H_4_-C_4_); 163.56, 170.55, 171.74 (pyrimidine-C_2,4,6_), (See Appendix A). **MS m/z (% relative abundance):** 386.15 (M+., 22.7%), 240.04 (100%) (See Appendix A). **Anal. Calcd.** (%) **for** C_21_H_18_N_6_O_2_ (386.41): C, 65.27; H, 4.70; N, 21.75. **Found** (%): C, 65.68; H, 5.03; N, 22.07.

#### 3.4.2. 2-(Benzo[d]oxazol-2-Ylmethylamino)-4-methoxy-6-(4-methoxyphenyl)pyrimidine-5-carbonitrile (**3b**)

Compound **3b** appeared to be an orange powder with 69% yield, having M.P. > 300 °C. **IR (KBr, cm^−1^)**: 3240(NH), 3022(C-H aromatic), 2940(C-H aliphatic), 2230 (CN), 1617(C=N), 1585 (C=C).**^1^H NMR (DMSO-*d_6_*, 400 MHz) δ:** 3.73 (s, 3H, C_6_H_4_-OCH_3_); 3.84 (s, 3H, pyrimidine-C_6_-OCH_3_); 4.41 (s, 2H, CH_2_); 6.47–6.52 (m, 2H, C_6_H_4_-C_3,5_-H); 6.65–6.71 (m, 2H, C_6_H_4_-C_2,6_-H); 7.14 (d, 2H, *J* = 8.44 Hz, benzo[*d*]oxazole-C_4,7_-H); 7.35 (d, 2H, *J* = 8.64 Hz, benzo[*d*]oxazole-C_5,6_-H); 12.73 (s, 1H, NH-CH_2_, D_2_O exchangeable), (See Appendix A). **^13^C NMR (DMSO-*d_6_*, 100 MHz) δ:** 56.20, 57.48 (2OCH_3_); 60.70 (CH_2_); 80.25 (pyrimidine-C_5_); 109.80 (benzo[*d*]oxazole-C_7_);114.02 (C_6_H_4_-C_3,5_); 115.78 (CN); 120, 124, 125 (benzo[*d*]oxazole-C_4,5,6_); 127.42 (C_6_H_4_-C_1_); 128.58 (C_6_H_4_-C_2,6_); 142.62, 151.78 (benzo[*d*]oxazole-C_3a,7a_); 153.28 (benzo[*d*]oxazole-C_2_); 160.25 (C_6_H_4_-C_4_); 165.57, 169.38, 172.30 (pyrimidine-C_2,4,6_), (See Appendix A). **Anal. Calcd.** (%) **for** C_21_H_17_N_5_O_3_ (387.39): C, 65.11; H, 4.42; N, 18.08. **Found** (%): C, 64.98; H, 4.14; N, 17.89.

#### 3.4.3. 2-(Benzo[d]thiazol-2-ylmethylamino)-4-methoxy-6-(4-methoxyphenyl)pyrimidine-5-carbonitrile (**3c**)

Compound **3c** appeared to be a beige powder with 77% yield, having M.P. > 300 °C. **IR (KBr, cm^−1^)**: 3210(NH), 3020(C-H aromatic), 2940(C-H aliphatic), 2230 (CN), 1622(C=N), 1595 (C=C).**^1^H NMR (DMSO-*d_6_*, 400 MHz) δ:** 3.89 (s, 3H, C_6_H_4_-OCH_3_); 4.21 (s, 3H, pyrimidine-C_6_-OCH_3_); 4.63 (s, 2H, CH_2_); 7.26-7.61 (m, 5H, C_6_H_4_-C_2,3,5,6_-H & benzo[*d*]thiazole-C_2_-H); 7.91-8.02 (m, 3H, benzo[*d*]thiazole-C_1,3,4_-H); 12.97 (s, 1H, NH-CH_2_, D_2_O exchangeable), (See Appendix A). **^13^C NMR (DMSO-*d_6_*, 100 MHz) δ:** 56.13, 57.52 (2OCH_3_); 61.56 (CH_2_); 80.76 (pyrimidine-C_5_); 117.43, 117.98 (C_6_H_4_-C_3,5_); 118.59 (CN); 120.95, 121.34, 124.92, 125.36 (benzo[*d*]thiazole-C_4,5,6,7_); 127.32 (C_6_H_4_-C_1_);128.01, 128.36 (C_6_H_4_-C_2,6_); 138.69, 152.24 (benzo[*d*]thiazole-C_3a,7a_); 162.08 (C_6_H_4_-C_4_); 169.16 (benzo[*d*]thiazole-C_2_); 166.16, 170.87, 171.69 (pyrimidine-C_2,4,6_), (See Appendix A). **Anal. Calcd.** (%) **for** C_21_H_17_N_5_O_2_S (403.46): C, 62.52; H, 4.25; N, 17.36. **Found** (%): C, 62.78; H, 4.46; N, 17.72.

#### 3.4.4. 2-(Benzo[b]thiophen-2-ylmethylamino)-4-methoxy-6-(4-methoxyphenyl)pyrimidine-5-carbonitrile (**3d**)

Compound **3d** appeared to be a chocolate-brown powder with 83% yield, having M.P. > 300 °C. **IR (KBr, cm^−1^)**: 3215(NH), 3022(C-H aromatic), 2937(C-H aliphatic), 2224 (CN), 1620(C=N), 1595 (C=C). **^1^H NMR (DMSO-*d_6_*, 400 MHz) δ:** 3.73 (s, 3H, C_6_H_4_-OCH_3_); 3.82 (s, 3H, pyrimidine-C_6_-OCH_3_); 4.02 (s, 2H, CH_2_); 6.47-6.53 (m, 6H, C_6_H_4_-C_2,3,5,6_-H & benzo[*b*]thiophene-C_5,6_-H); 6.88 (s, 1H, benzo[*b*]thiophene-C_3_-H); 7.17 (d, 2H, *J* = 8.44 Hz, benzo[*b*]thiophene-C_4,7_-H); 12.36 (s, 1H, NH-CH_2_, D_2_O exchangeable), (See Appendix A). **^13^C NMR (DMSO-*d_6_*, 100 MHz) δ:** 56, 57 (2OCH_3_); 60.24 (CH_2_); 81.74 (pyrimidine-C_5_); 116.50 (C_6_H_4_-C_3,5_); 117 (CN); 122.74, 123.55, 123.72, 123.72, 124.53 (benzo[*b*]thiophene-C_3,4,5,6,7_); 127.36 (C_6_H_4_-C_1_); 128.50, 128.57 (C_6_H_4_-C_2,6_); 138.54 (benzo[*b*]thiophene-C_3a,7a_); 141.76 (benzo[*b*]thiophene-C_2_); 161.73 (C_6_H_4_-C_4_); 164.53, 170.03, 172.74 (pyrimidine-C_2,4,6_), (See Appendix A). **Anal. Calcd.** (%) **for** C_22_H_18_N_4_O_2_S (402.47): C, 65.65; H, 4.51; N, 13.92. **Found** (%): C, 65.77; H, 4.81; N, 14.09.

### 3.5. General Procedure for Synthesis of Compounds ***5a***–***5d***

Compound (**1**) (3 mmol) was fused with an equimolar quantity of various sulfonyl chloride derivatives, namely, benzenesulfonyl chloride, 4-methylbenzene-1-sulfonyl chloride, 4-*tert*-butylbenzene-1-sulfonyl chloride, and 4-(trifluoromethyl)benzene-1-sulfonyl chloride at ~200 °C for 2 h. Later, after cooling, the mixture was triturated with ethanol and then poured onto crushed ice. The synthesized compound was then filtered and rinsed many times with water to remove impurities. Following that, the product was dried and crystallized in ethanol.

#### 3.5.1. *N*-(5-Cyano-4-methoxy-6-(4-methoxyphenyl)pyrimidin-2-yl)benzenesulfonamide (**5a**)

Compound **5a** appeared to be a white-colored powder with 81% yield, having M.P. 241–243 °C. **IR (KBr, cm^−1^)**: 3240(NH), 3060(C-H aromatic), 2922(C-H aliphatic), 2230 (CN), 1637(C=N), 1583 (C=C). **^1^H NMR (DMSO-*d_6_*, 400 MHz) δ:** 3.78 (s, 3H, C_6_H_4_-OCH_3_); 4.22 (s, 3H, pyrimidine-C_6_-OCH_3_); 7.37-7.39 (m, 4H, CH_3_O-C_6_H_4_-C_2,3,5,6_-H); 7.58-7.63 (m, 5H, SO_2_-C_6_H_5_-H); 11.75 (s, 1H, NH, D_2_O exchangeable), (See Appendix A). **^13^C NMR (DMSO-*d_6_*, 100 MHz) δ:** 57.48, 58.57 (2OCH_3_); 81.25 (pyrimidine-C_5_); 117.80 (CH_3_O-C_6_H_4_-C_3,5_); 118.20 (CN); 127.42, 128.58, 129.31, 134.02, 145.87 (CH_3_O-C_6_H_4_-C_1,2,6_ & SO_2_-C_6_H_5_); 160.25 (CH_3_O-C_6_H_4_-C_4_); 170.38, 171.50, 171.70 (pyrimidine-C_2,4,6_), (See Appendix A). **MS m/z (% relative abundance):** 396.09 (M**^+.^**, 20.5%), 76.04 (100%) (See Appendix A). **Anal. Calcd.** (%) **for** C_19_H_16_N_4_O_4_S (396.42): C, 57.57; H, 4.07; N, 14.13. **Found** (%): C, 58.04; H, 3.97; N, 13.88.

#### 3.5.2. *N*-(5-Cyano-4-methoxy-6-(4-methoxyphenyl)pyrimidin-2-yl)-4-methylbenzene-sulfonamide (**5b**)

Compound **5b** appeared to be a buff-colored powder with 78% yield, having M.P. 283–285 °C. **IR (KBr, cm^−1^)**: 3235(NH), 3053(C-H aromatic), 2922(C-H aliphatic), 2222 (CN), 1632(C=N), 1585 (C=C).**^1^H NMR (DMSO-*d_6_*, 400 MHz) δ:** 2.29 (s, 3H, CH_3_); 3.74 (s, 3H, C_6_H_4_-OCH_3_); 3.97 (s, 3H, pyrimidine-C_6_-OCH_3_); 6.45–6.48 (m, 4H, CH_3_O-C_6_H_4_-C_2,3,5,6_-H); 6.54 (d, 2H, *J* = 8.36 Hz, SO_2_-C_6_H_4_-C_3,5_-H); 6.93 (d, 2H, *J* = 8.44 Hz, SO_2_-C_6_H_4_-C_2,6_-H); 11.18 (s, 1H, NH, D_2_O exchangeable), (See Appendix A). **^13^C NMR (DMSO-*d_6_*, 100 MHz) δ:** 23.53 (CH_3_); 57.41, 58.10 (2OCH_3_); 81.97 (pyrimidine-C_5_); 117.93 (CH_3_O-C_6_H_4_-C_3,5_); 118.40 (CN); 127.69 (CH_3_O-C_6_H_4_-C_1_), 128.33, 128.49 (SO_2_-C_6_H_4_-C_2,6_-H); 128.58, 128.86 (CH_3_O-C_6_H_4_-C_2,6_); 129.02,129.42 (SO_2_-C_6_H_4_-C_3,5_-H); 138.58, 139.34 (SO_2_-C_6_H_4_-C_1,4_-H); 160.19 (CH_3_O-C_6_H_4_-C_4_); 170.74, 171.25, 172.90 (pyrimidine-C_2,4,6_), (See Appendix A). **Anal. Calcd.** (%) **for** C_20_H_18_N_4_O_4_S (410.45): C, 58.53; H, 4.42; N, 13.65. **Found** (%): C, 58.14; H, 4.18; N, 13.84.

#### 3.5.3. 4-*Tert*-Butyl-*N*-(5-cyano-4-methoxy-6-(4-methoxyphenyl)pyrimidin-2-yl)benzene sulfonamide (**5c**)

Compound **5c** appeared to be a yellow powder with 72% yield, having M.P. 276–278 °C. **IR (KBr, cm^−1^)**: 3190(NH), 3010(C-H aromatic), 2930(C-H aliphatic), 2240 (CN), 1618(C=N), 1579 (C=C).**^1^H NMR (DMSO-*d_6_*, 400 MHz) δ:** 1.28 (s, 9H, C(CH_3_)_3_); 3.81 (s, 3H, C_6_H_4_-OCH_3_); 4.24 (s, 3H, pyrimidine-C_6_-OCH_3_); 7.39 (d, 2H, *J* = 6.88 Hz, CH_3_O-C_6_H_4_-C_3,5_-H); 7.58–7.64 (m, 4H, CH_3_O-C_6_H_4_-C_2,6_-H & SO_2_-C_6_H_4_-C_3,5_-H); 7.93 (d, 2H, *J* = 8.44 Hz, SO_2_-C_6_H_4_-C_2,6_-H); 11.70 (s, 1H, NH, D_2_O exchangeable), (See Appendix A). **^13^C NMR (DMSO-*d_6_*, 100 MHz) δ:** 31.92 (C(CH_3_)_3_); 34.51 (C(CH_3_)_3_); 57.15, 58.28 (2OCH_3_); 80.61 (pyrimidine-C_5_); 117.71 (CH_3_O-C_6_H_4_-C_3,5_); 118.00 (CN); 125.58 (SO_2_-C_6_H_4_-C_3,5_-H); 128.53, 128.57 (CH_3_O-C_6_H_4_-C_2,6_); 129.31, 129.36 (SO_2_-C_6_H_4_-C_2,6_-H); 130.30 (CH_3_O-C_6_H_4_-C_1_), 137.87, 154.63 (SO_2_-C_6_H_4_-C_1,4_-H); 160.62 (CH_3_O-C_6_H_4_-C_4_); 170.88, 171.19, 172.29 (pyrimidine-C_2,4,6_), (See Appendix A). **Anal. Calcd.** (%) **for** C_23_H_24_N_4_O_4_S (452.53): C, 61.05; H, 5.35; N, 12.38. **Found** (%): C, 60.95; H, 5.19; N, 12.12.

#### 3.5.4. *N*-(5-Cyano-4-methoxy-6-(4-methoxyphenyl)pyrimidin-2-yl)-4-(trifluoromethyl)benzene sulfonamide (**5d**)

Compound **5d** appeared to be a yellow-colored powder with 75% yield, having M.P. 291–293 °C. **IR (KBr, cm^−1^)**: 3232 (NH), 3045(C-H aromatic), 2920(C-H aliphatic), 2230 (CN), 1627(C=N), 1585 (C=C). **^1^H NMR (DMSO-*d_6_*, 400 MHz) δ:** 3.87 (s, 3H, C_6_H_4_-OCH_3_); 4.18 (s, 3H, pyrimidine-C_6_-OCH_3_); 7.45-7.47 (m, 4H, CH_3_O-C_6_H_4_-C_2,3,5,6_-H); 7.54 (d, 2H, *J* = 8.28 Hz, SO_2_-C_6_H_4_-C_3,5_-H); 7.93 (d, 2H, *J* = 8.44 Hz, SO_2_-C_6_H_4_-C_2,6_-H); 11.88 (s, 1H, NH, D_2_O exchangeable), (See Appendix A). **^13^C NMR (DMSO-*d_6_*, 100 MHz) δ:** 56.96, 57.67 (2OCH_3_); 81.03 (pyrimidine-C_5_); 117.86 (CH_3_O-C_6_H_4_-C_3,5_); 118.00 (CN); 120.98 (CF_3_); 124.89, 125.42 (SO_2_-C_6_H_4_-C_3,5_-H); 127.42 (CH_3_O-C_6_H_4_-C_1_), 128.80 (CH_3_O-C_6_H_4_-C_2,6_); 129.28, 129.35 (SO_2_-C_6_H_4_-C_2,6_-H); 138.58, 145.54 (SO_2_-C_6_H_4_-C_1,4_-H); 159.07 (CH_3_O-C_6_H_4_-C_4_); 170.09, 171.89, 172.61 (pyrimidine-C_2,4,6_), (See Appendix A). **Anal. Calcd.** (%) **for** C_20_H_15_F_3_N_4_O_4_S (464.42): C, 51.72; H, 3.26; N, 12.06. **Found** (%): C, 52.03; H, 3.47; N, 11.89.

### 3.6. Cell Lines

Breast carcinoma (MCF-7), renal cell carcinoma (A498), lung cancer (A549), and liver carcinoma (HepG2) cell lines, in addition to non-tumorigenic WI-38 cell lines, were collected from the American Type Culture Collection. The cells were grown in DMEM (Invitrogen/Life Technologies) containing 1% penicillin-streptomycin, 10 µg/mL of insulin (Sigma), and 10% FBS (Hyclone).

### 3.7. Cytotoxicity Protocol

To check the possible cellular toxicity, synthesized compounds were subjected to the MTT [3-(4,5-dimethylthiazol-2-yl)-2,5-diphenyltetrazolium bromide] assay against the MCF7, A549, A498, and HepG2 cell lines. Treatment of cells was performed when the obtained cell lines were grown in a respectable medium and rinsed with Trypsin 0.25% (*w*/*v*) and EDTA (0.53 mM) solution to remove the Trypsin inhibitor. Later, 2.0–3.0 mL of Trypsin EDTA solution was added into the flask containing the cell layers for 5–15 min to disperse the cell layer, which was observed through an inverted microscope. By gently pipetting, aspirated cells were added into a 6.0–8.0 mL growth medium. The cell suspension, along with the medium and the treated cells, was centrifuged at approximately 125× *g* for 5 to 10 min. Again, the cell pellet was inoculated in a fresh growth medium for incubation at 37 °C for 24 h. Later, treated cells (100 µL) having a cellular density of 1.2–1.8 × 10,000 cells/well were incubated with the series of concentrations of the synthesized compounds (100 µL) for 24 h at 37 °C in 96-well plates. An amount of 20 μL of a 5-mg/mL MTT solution was applied after a 24-h drug-treatment period, and it was incubated for 4 h. To dissolve the produced purple formazan, each well received 100 μL of dimethyl sulfoxide (DMSO). The obtained plates were observed with the inverted microscope, and the MTT assay proceeded by following the reported procedure [43].

### 3.8. Cell Cycle Analysis

Compound **5d** was applied to MCF-7 cell culture for 48 h at its IC_50_ concentration. Briefly, cells were rinsed twice with phosphate buffer saline (PBS) (ice-cold) and then centrifuged and/or fixed in 70% (*v*/*v*) ethanol (ice-cold). The solution was rinsed with PBS, suspended with RNase (0.1 mg/mL), stained with propidium iodide (PI) (40 mg/mL), and analyzed via flow cytometry utilizing the FACS Calibur (Becton Dickinson, Franklin Lakes, NJ, USA). Through Cell-Quest software (Becton Dickinson), cell cycle distributions were calculated.

### 3.9. Apoptosis Assay

The MCF-7 cell culture was treated with an IC_50_ of compound **5d** for 48 h. The cancer cells were then suspended in PBS, centrifuged, and preserved in 99% ethanol (ice-cold). Cells suspended in ethanol were centrifuged for 5 min before being suspended in PBS and re-centrifuged once again. The supernatant was decanted, and the cells were stained at 37 °C with annexin V fluorescein isothiocyanate and PI using the apoptosis detection kit (BD Biosciences, San Jose, CA, USA) following the manufacturer’s instructions. A flow cytometer was used to examine the binding of annexin V-FITC and PI. The FACS Calibur was used to assess the cells, and Phoenix Flow Systems and Verity Software House were employed to calculate their distributions.

### 3.10. COX-2 Inhibition Assay

Fluorometric inhibitor screening kits (Biovision, Zurich, Switzerland) were utilized for the COX-2 enzyme inhibition test, as described before [44]. The inhibition percentages of pyrimidine derivatives were evaluated in the concentration range of 10^−5^ M and 10^−9^ M. The reference agents, i.e., Celecoxib and Nimesulide, were also used in the same range of concentrations.

### 3.11. Statistical Analysis

The results were stated as mean ± SD. Experiments were performed in triplicate at a minimum. One-way analysis of variance (ANOVA) was applied to compare data sets, whereas a *p*-value < 0.05 was described as significant.

### 3.12. Docking Studies

All docking experiments were performed for the target structures using Molecular Operating Environment software (MOE 2014, https://www.chemcomp.com/Products.htm) (accessed on 15 March 2022) to evaluate the free energy of binding and to explore the binding mode toward the COX-2 enzyme. The experiments used proteins retrieved from the Protein Databank (PDB: 1CX2, Resolution: 3.00 Å (https://www.rcsb.org/structure/1CX2) (accessed on 15 March 2022) and considered as a target for docking simulation. First, the crystal structure of the protein was prepared by removing water molecules while retaining the essential chain and the ligand, Celecoxib. After that, the protein was protonated, the energy was minimized, and the binding pocket of the protein was defined. The 3D structures of the target compounds were sketched using Chem3D 15.0 and saved in MDL molfile format after minimizing the energy. Molecular docking of the final target compounds was performed using a default protocol against the target receptor. In each case, 30 docked structures were generated using genetic algorithm searches; London dG and GBVI/WSA dG were used for scoring 1 and 2, respectively.

## 4. Conclusions

In conclusion, an effective and environmentally benign design and synthesis of new hybrids of 6-(4-methoxyphenyl)-5-carbonitrile-based pyrimidine bearing either benzo[*d*]imidazole, benzo[*d*]oxazole, benzo[*d*]thiazole, and benzo[*b*]thiophene derivatives via methylene amino linker **3a**–**3d** (general formula A) or various sulphonamide phenyl moieties **5a**–**5d** (general formula B) at the C-2 position was achieved. All the designed compounds were screened for their COX-2 inhibition. Almost all of the compounds displayed good COX-2 inhibition activity, comparable to Celecoxib and higher than Nimesulide. However, the pyrimidine substituted with various sulphonamide phenyl derivatives **5a**–**5d** (general formula B) showed higher inhibition of COX-2 enzyme activity than the pyrimidines bearing benzimidazole, benzoxazole, benzothiazole, and benzothiophene derivatives **3a**–**3d** (general formula A). Among these series, the most potent agents (**3b**, **5b**, and **5d**) were further assessed for their anticancer activity. The three compounds demonstrated potent cytotoxic activity against four cell lines, i.e., MCF7, A549, A498, and HepG2, and the activity was comparable to or 2- to 13-fold higher than doxorubicin, with low cytotoxicity on the normal WI-38 cell line. The effect of compound **5d** on cell cycle progression and apoptosis induction was investigated, and it was found that compound **5d** could seize cell growth at the sub-G1 and G2/M phases, as well as increase the proportion of early and late apoptotic rates in MCF-7 cells by nearly 13- and 60-fold, respectively. Moreover, in silico studies for compounds **3b**, **5b**, and **5d** revealed promising findings, such as strong GIT absorption, an absence of BBB permeability, nil-to-low drug–drug interactions, good oral bioavailability, and optimal physicochemical properties, indicating their potential as promising therapeutic candidates. In summary, the activity of the proposed compounds, namely **3b**, **5b**, and **5d**, was presented as a lead for further investigating their potential to serve as multi-target compounds in the fields of both COX-2 inhibition and antiproliferative activities, which will be discussed in future articles.

## Data Availability

Not applicable.

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
