# Peer review of "New Pyrimidine-5-Carbonitriles as COX-2 Inhibitors: Design, Synthesis, Anticancer Screening, Molecular Docking, and In Silico ADME Profile Studies"

_molecules, 2022, doi:10.3390/molecules27217485_

Round 1

Reviewer 1 Report

Dear all,

The manuscript entitled “New pyrimidine-5-carbonitriles as COX-2 inhibitors: Design, synthesis, anticancer screening, molecular docking and in silico ADME profile studies” presents a well-put-together medicinal chemistry study where 8 novel pyrimidine-based compounds were designed and synthesized according to new, green methodologies. Due to their COX-2 inhibiting and antiproliferative activities, these compounds were shown to be promising leads for further development as anticancer agents with potential multitarget activities, particularly those exhibiting favorable in silico ADME profiles, which were also reported in the manuscript. I am for the publication of this work after a few minor comments are addressed:

1. Line 18: Please replace “of 10-8 M” with “, with IC50 values in the submicromolar range”. 

2. Line 27: Please replace “folds” with “fold”. This is a recurring issue throughout the manuscript (e.g. line 57 “kinases” instead of “kinase”, line 60 “antioxidants” instead of “antioxidant”, and line 251 again “folds” instead of “fold”).

3. Line 35: Please replace “is a member of” with “are members of”.

4. Lines 47-48: When referring to “in multiple drug resistance”, did the authors mean “in drug-resistant tumors”? Please adjust accordingly.

5. Line 52: Please replace the word “essential” with “important”.

6. Line 54: Please make a paragraph after “...and metastasis of cancer (14).”

7. Line 56: Please replace “Literature” with “literature”.

8. Line 57: Please provide more references besides reference (15). Only one reference does not make it “evident”. Alternatively, replace the word “evident” with “suggested” or an equivalent term.

9. Line 59: Please add “of acting as anticancer, antioxidant and COX-2 inhibiting agents” after “possess the capability”.

10. Lines 60-61: “Please replace “multiple tasking” with “multitarget”.

11. Line 61: Please replace “is always beneficial” with “may be beneficial”.

12. Line 62: Please replace the word “Besides” with “In addition,” or an equivalent expression.

13. Line 63: Please replace the word “incredible” with a more suitable expression (e.g. notable).

14. Line 79: Please remove the quotes before and after -NH.

15. Line 84: Please remove the additional space after the = signs.

16. Lines 77-87: Please provide IC50 values for all described compounds for consistency (right now only a few are described).

17. Figure 1: Please adjust the figure so that the aromatic rings do not look vertically “stretched”. Also, please add the appropriate references and correct the sentences to “Reported cyanopyrimidine-containing anticancer agents in the literature” and “Cyanopyrimidine bearing a p-methoxyphenyl moiety as selective COX-2 inhibitors”.

18. Line 91: The sentence should start with “The…”. There is a lack of the article “the” throughout this paragraph, please adjust.

19. Lines 92-93: The sentence is confusing, please rephrase. It is not the pyrimidine-based sulfonamide phenyl pharmacophore exhibiting more potency than celecoxib, it is the whole compound 7. Also, the expression “i.e.” means something like “in other words”, which does not fit the sentence. 

20. Line 93: Please cite Figure 2 after compound 7. The same for the remaining compounds referred to in this paragraph. 

21. Line 97: The word “portraying” should be replaced with “suggesting” or “indicating”.

22. Line 99: There is a typo in “benzoXazoles”. Also, the word “chromophore” seems to be here by mistake. Did the authors mean “pharmacophore”? Please rephrase.

23. Line 102: Same issue as in my comment 19. “...structures with the pyrimidine scaffold” should be followed by something like “such as in compound 9…”.

24. Lines 112-113 and throughout the manuscript: The numbering 1-10 had already been established for the compounds presented in Figures 1 and 2. It seems inadequate to restart the numbering for the new compounds reported by the authors. Please correct throughout.

25. Figures 1 and 2: Since the target compounds were designed based on compounds both in Figure 1 and Figure 2, it would be more adequate to have them all in one single Figure. Otherwise, it seems that only 7-10 were the starting points and, on the other hand, it is not clear where the cyano and the p-methoxyphenyl groups came from. Figures need to be self-explanatory. Also, please add references to the sentences in Figure 2 and correct the typo in “benzoXazole”.

26. Line 124: “A solvent-free…” should read instead of “The solvent-free…”.

27. Line 132: “the green strategy” should be replaced with “a much greener” or “much more environmentally friendly” or an equivalent expression. It is being compared to other strategies after all. 

28. Lines 150-151: Please correct to “The remaining expected proton and carbon signals were detected in the 1H NMR and 13C NMR spectra, and are attributed in the experimental section”.

29. Scheme 1 and Scheme 2: Please provide yields for each described compound.

30. Line 164: Please replace “Compound 5b structure” with “The structure of compound 5b”.

31. Line 173: Please replace “formula” with “structure”.

32. Line 174: Please replace “signifying” with “suggesting”.

33. Line 185: “...remarkable increase in activity…” compared to which compound? Please clarify.

34. Lines 203-204: I have an issue with this sentence. The IC50 is in the same order of magnitude for all compounds. Also, 3a and 3c are the ones with the highest % of COX-2 inhibition at the lowest concentrations. Statistical significance would be required to support this sentence. Please provide statistical data, namely p-values for all comparisons after ANOVA.

35. Line 215: Please replace “...interacted with this cell line” with “...were active in this cell line” or an equivalent expression.

36. Line 218: Please add spaces before and after the = sign for consistency.

37. Line 229: “...were safe…” This is too strong for the data presented in the manuscript. Please rephrase by referring to the determined therapeutic window. Also, please compare it with the one determined for doxorubicin in the same experimental conditions.

38. Tables 1-4 and Figures 3 and 4: Please provide statistical data.

39. Line 235: “The most active compound…” Please present statistical data supporting this claim.

40. Line 242: “...and the subsequent induction of apoptosis”. Please remove it, it seems that the authors cannot conclude this based on the data presented to this point.

41. Table 4 and Figure 4: It should be clear the presented values are cell counts. It is not mentioned anywhere as far as I could tell apart from the experimental section.

42. Throughout section 2.3. Please fix the Å symbol.

43. Line 270: “binds with” should be replaced with “binds to”.

44. Line 325: Please replace “implying” with “suggesting”. Also, is the MW-based observation based on Lipinsky’s rule of 5? Please clarify.

45. Line 350: Please replace the word “satisfactory” with “reliable” or an equivalent expression.

46. S7: There seem to be way more 13C signals in the spectrum than there are carbon atoms in the molecule, please double-check.

47. Line 525: The expression “interacted with” seems rather off. Please rephrase. Also, there needs to be a mention of the vehicle used in these experiments. Based on the solubility of the compounds I am assuming DMSO. In any case, what was the final % of vehicle used in these assays? Also, please add the vehicle control data or state that no cytotoxicity was observed with x% of vehicle.

48. Line 555: No p-values were described in the paper. Please adjust as per my previous comments.

49. Lines 572-576: This sentence is poorly written. It was not “The method for the synthesis” being “designed and synthesized”. Please rephrase.

50. If I understood correctly, the authors believe that these compounds are COX-2 inhibitors AND antiproliferative agents, and not antiproliferative agents DUE TO being COX-2 inhibitors. If so, I feel that the potential of these multitarget compounds should be further discussed with regard to how they would be valuable and innovative anticancer treatments for gathering COX-2 inhibiting and antiproliferative activities in one single molecule. This should be explored in the discussion and/or conclusion sections.

51. Line 595: Please replace the word “due” with “future”.

Thank you.

Reviewer 2 Report

1. There is some spelling and grammatic mistakes throughout the manuscript, so I highly recommend to revise this manuscript in a certified translation office.

2. The rational design is unspecified and should be more clarified.

3. SAR explanation is poorly written. 

Reviewer 3 Report

In this manuscript, the authors presented their work on a synthesis of new pyrimidine-5-carbonitriles using a green strategy due to the solvent-free conditions, easily obtainable resources, and high yields with facile product isolation. The most active pyrimidine derivatives presented COX-2 percent inhibition and IC50 values that are nearly equal to Celecoxib and approximately 4.7-, 9.3-, and 10.5-fold higher than Nimesulide and also demonstrated anticancer activity comparable to or better than doxorubicin against four MCF-7, A549, A498, and HepG2 cell lines, with IC50 values in nanomolar in addition to low cytotoxicity on the normal W38-I cell line. The Mechanistic study was investigated for the most active compound and it was proven that cytotoxicity happens by the arrest of cell growth at the G2/M phase and the subsequent induction of apoptosis.

In general, the manuscript is very concise, well-organized, and written and the compounds are well-characterized. The authors found a green protocol for obtaining the compounds, and the biological assays provided a lot of information. On this basis, I recommend publication in Molecules provided that the authors will address in a revised version some points, described below:

Lines 54-57:  The authors describe that the pyrimidine nucleus is among the major structural cores in medicinal chemistry and it is evident from the Literature that pyrimidine and its derivative compounds hold kinases and angiogenesis inhibition potential. There is a strong argument for only one reference cited, the authors should provide more references to reinforce this argument.

Structure 3 of figure 1 contains the wrong geometry of the alkyne group, please redraw in a linear geometry

The rationale is concise, but it is missing references from Claudiu T. Supuran about the role of sulphonamides moiety for inhibition of the enzyme COX-2.
